# Metadata Schema for Folktales in the Mekong River Basin

**Kanyarat Kwiecien** [1], **Wirapong Chansanam** [1], **Thepchai Supnithi** [2], **Jaturong Chitiyaphol** [1]
**and Kulthida Tuamsuk** [1,*]

1. Department of Information Science, Faculty of Humanities and Social Sciences, Khon Kaen University, Khon Kaen 40002, Thailand; kandad@kku.ac.th (K.K.); wirach@kku.ac.th (W.C.); jaturongc@kkumail.com (J.C.)
2. National Electronics and Computer Technology Center, Bangkok 12120, Thailand; thepchai@nectec.or.th
* Correspondence: kultua@kku.ac.th; Tel.: +66-89-8614696

**Abstract:** The aim of this study was to analyze the content, context, and structure of folktales from the Mekong River Basin, and to develop a metadata schema for data description and folktale storage. The research was conducted using the MAAT metadata lifecycle model, which comprises the following four steps: (1) conducting an information content analysis; (2) creating metadata requirements, (3) developing a metadata schema; and (4) carrying out a metadata service and evaluation. The folktale analysis, based on Anne Gilliland's information object analysis, revealed the following: (1) the folktale content consists of types of tales, and the morals, beliefs, and parts they incorporate; (2) the folktale context consists of and names distributors, characters, scenes, magical objects, ethnic groups, languages, countries, relationships between tales, and their sources; (3) the folktale structure includes verbal, non-verbal, and mixed forms. The metadata schema development adopted the functional requirements for bibliographic records concepts and existing metadata standards, resulting in metadata with the following 18 elements: identifier, title, creator, contributor, description, relation, language, medium, sources, date, rights, keyword, character, moral, ethnic group, motif, place, and country. The metadata elements were described using the categories: name, definition, format, example, and note.

**Keywords:** metadata schema; folktale metadata; Mekong River Basin; information object analysis; functional requirements for bibliographic records



## 1. Introduction

The Mekong River Basin (MRB) is the cultural center of the countries that border the Mekong River, namely, Thailand, Laos, Cambodia, Myanmar, Vietnam, and the People's Republic of China. Although this region is not very large, it is rich in cultural heritage and wisdom. For its intellectual and cultural heritage, which comprises eighteen tangible cultural-heritage items and six intangible cultural-heritage items [1], the Mekong Region has been registered as a World Cultural Heritage site by UNESCO. The cultures of the Mekong countries are similar because they share the Mekong River and have similar cultural roots. The people of this region share a similar way of life, religious beliefs, and characteristics. The phrase, "the common cultures of MRB" [2] was coined to express this heritage. For instance, the cultures of this region have common linguistic roots and similar music, dance, traditions, and performing arts. The Mekong River Basin is thus one of the most important cultural heritage sites in Asia and the world.

The folktale is a kind of intangible cultural heritage. The term refers to stories that have been passed down, either from oral to written methods or vice versa. When folktale is associated with specific localities, its content is adapted to fit the society, way of life, and the environment of that locality. The details of stories differ, reflecting the influence of each locality [3]. As folktale contains the local wisdom, beliefs, values, norms, and other characteristics of particular communities, it is a tool that can be used to understand cultures and the way people lived in the past. Consequently, it is crucial to preserve

intangible cultural knowledge, especially folktales and other knowledge. This is evident from the many studies that have used the digital humanities method to preserve folktales worldwide. Examples include the knowledge technologies used to describe the semantics of Bulgarian folktale heritage [4], and the ontology-based search and document retrieval in a digital folk-songs library [5]. Regarding research on cultural heritage, there are several researchers in Asian countries, such as [6], who developed a multimedia system that educates children through storytelling; [7], who developed the metadata standard for describing Chinese local music; and Chansanam and [8], who developed an intangible cultural-heritage knowledge framework for ontology development.

The folktales of countries in the Mekong Region are presented and passed down in several forms. These can be grouped by presentation style into three categories: (1) verbal knowledge, or knowledge related to words, such as local music, and riddles; (2) nonverbal knowledge, in which content is passed down without the use of words, through art, crafts, and architecture; and (3) a mix of verbal and nonverbal knowledge, including performances and rituals. Research has also shown knowledge management for folktales from the Mekong region which can be broken down into sub-categories, such as myths, fairy tales, and legends. The content of folktales within the Mekong Region mainly concerns the origin of the Mekong River and the legends and beliefs of people who lived along the river, such as the tale of the Sa Khu Lu giant in parts of China, Myanmar, Lao, Thailand, Cambodia, and Vietnam. Because the giant was huge, it created a loud sound similar to thunder or an earthquake whenever it moved, splitting the ground into a waterway and creating the Mekong River. There is also a myth about Naga or big snakes, which can be found in the folktales of many countries [9].

To help researchers learn about the similar histories and cultures of people within the Mekong Region, tools or platforms are needed to manage cultural-heritage information in a standardized way such that folktales from different countries can be shared. Research on platforms designed to manage cultural-heritage knowledge at an international standard show that many countries are using standard platforms. One of these is Europeana (https://www.europeana.eu/en (accessed on 5 April 2021)), a digital collection for managing the cultural heritage of the EU from pre-historical times up to the present. This platform stores and provides access to information, which can be searched using content, colors, timeframe, the names of related people, and sources of information [10]. The World Digital Library (WDL) (https://www.wdl.org/en/, accessed on 5 April 2021) is another example of a digital platform that can be used to manage cultural-heritage information; it was developed by the United States Library of Congress. The WDL users can access important historical evidence from different sources within one platform. The stored content includes books, original handwritten manuscripts, maps, newspapers, magazines, publications, photographs, voice recordings, and films. Content within the WDL can be searched using timeframes, content, language, and other features [11].

Developing an effective digital platform for semantic information searches requires a point of access to informational content, where search terms can be analyzed and managed. The designated entry point must describe the characteristics of information objects, ensuring that researchers can gain access to the information efficiently. Studies of the knowledge management of folktale from the Mekong Region include [9], which analyzed the scope of folktales in the Mekong Region, and [3], which developed a folktale ontology for the region. However, efforts to develop a digital platform for semantic searches of folktales in this region continue to be hindered by a lack of metadata to describe their characteristics. To develop a complete and efficient semantic search, it is essential to develop metadata.

Metadata refers to the structural descriptive or narrative information associated with information resources, information objects, or information systems; it allows this information to be managed, searched, and used [12]. A description of informational resources must include three key features of the information resources: (1) content, which refers to the story, information, or items recorded within the resources. These are also latent things that cannot be seen, except through analysis and interpretation; (2) context, which refers

to the basic information that identifies the who, what, where, and how the document is created. It provides information about the creation, preservation, and implementation of the information, as well as copyright protection details; and (3) structure, which refers to the external form of resources and their content management [13–15].

A literature review on the development of metadata schema to manage cultural-heritage information revealed various types of metadata, including metadata for managing ancient manuscripts, such as the palm-leaf manuscript [16], an ancient codex and Samut Khoi [17], images of the Buddha [18], inscriptions [19], and painted murals [20]. Researchers who investigated the development of a platform to manage cultural heritage included [21], who developed the project to store cultural heritage, carried out by the European Union. This system, the Europeana Project, allows people to gain access to European art and culture via the Internet. It includes information about more than four million images and audio and video entries. The study also developed a metadata standard, which used information provided by the owner of each image to explain it, based on the content of the image or stored document. Kress [22] developed a database and website to preserve murals through a project initiated by the Research Group for Baroque Ceiling Painting in Central Europe (BCPCE), a group of members from various countries, including Austria, Croatia, the Czech Republic, Germany, Hungary, Italy, Poland, Slovakia, Slovenia, and the United States. The group created a gallery of ancient and modern paintings by building a database and website to preserve them and developed metadata to manage the gallery; this was achieved by focusing on the iconography or interpreting the content of the paintings. Their explanations help to describe the stories within the paintings and to link them to other paintings. The digital content is searchable by categories related to the content of the paintings.

It is clear that the existing metadata standards were developed to manage different types of information resources, reflecting physical differences, varied content, and user needs [23]. Most metadata standards describe information related to items in libraries. Metadata standards rarely focus on describing and linking the appropriate semantic data to folktales. In addition, the existing platforms are managed at the organizational level. This research, therefore, develops metadata for folktales since it contains many different specific elements of data that are different from the other information resources. Folktales in the Mekong region were used to identify the data elements in this research because the developed metadata can be used for linking the folktales which share some common cultural information in the region, such as ethnic groups, languages, morals, beliefs, etc., which are rarely found in the existing metadata standards. In addition, the metadata of folktales in the Mekong region can be further used for managing folktales in the other region.

Folktale knowledge is not always recorded or kept within organizations. It can also be held by various individuals; this is called tacit knowledge. As most folktales exist in oral forms, knowledge can be hidden within academics, local sages, and experts. A new metadata development that can manage folktale knowledge will help to reduce the limitations caused by different languages and physical characteristics and provide an opportunity for people who are interested in folktale in the Mekong Region to collaborate in storing, linking, accessing, and exchanging that knowledge. Such an advancement will benefit the sustainable economy and social developments in the future.

For the reasons discussed above, the researcher considered it important to accumulate written and oral folktales from the Mekong Region, and then to analyze this knowledge and its various attributes. The analysis will help to design a metadata schema that can fully describe the knowledge and attributes of these folktales, while remaining consistent with the usage context. This can be done by using the functional requirements for bibliographic records (FRBR) [24] to develop a conceptual model, which can be used to develop a future semantic search system for folktales. This research and development will help both organizations and individuals in the Mekong region to store, preserve, and access the meaning of folktales and to use folktale knowledge in different forms. The results can be used to

improve digital information storage and digital libraries, as well as to support research designed to create products, services, and new innovations that can advance the sustainable economic and social development of Mekong Region countries.

## 2. Research Objectives

The research objectives are to analyze attributes in the content, context, and structure of folktales, and to develop a metadata schema to manage folktales in the Mekong River Basin.

## 3. Method and Results

The metadata schema was developed by adopting the Metadata Architecture and Application Team's [25] metadata life cycle model, which consists of the following four steps: (1) analyzing the information content, (2) creating the metadata requirements, (3) developing the metadata schema, and (4) developing the metadata-service system and evaluation.

### 3.1. Information Analysis

A content analysis was used to determine the function and elements of the metadata by analyzing folktale content that was recorded and distributed in several forms, including books, databases, video clips, songs, stories, and public-relations messaging (430 titles in total), distributed among the six Mekong River Basin countries. The analysis followed [26]'s principle of information object analysis, in which information objects consist of three parts: content, context, and structure. The researcher had previously studied related metadata standards, including the Dublin core metadata element set [27]; the metadata object description schema(MODs) [28]; the Visual Resource Association (VRA Core) [29], used to describe visual arts, such as paintings, sculptures, and architecture; ategories for the description of works of art (CDWA) [30], used to explain the arts, architecture, and cultural objects; and performance arts metadata [31]. These metadata provided guidelines for analyzing and developing a metadata schema to describe folktale knowledge in the Mekong Region.

The analysis of folktale, conducted in accordance with [26] information object analysis, produced the following results:

### 3.1.1. Content

Content is the data which are analyzed and interpreted from the story within the resources. The content of the folktales can be classified into four elements as follows.

(1)　The types of folktale can be grouped into eight categories, based on content:
- Novel and romantic tales are stories about love; in most of these, love must overcome obstacles and it leaves traces in various locations. Examples include The Legend of Khun Nang Non, Ma Mia, and Noi Jai Ya (a Lanna tale);
- Hero tales are stories about battles, either fictional or based on true stories. The protagonist is a hero, who has an adventure to take back land, a lover, or magical items. Sin Sai (an Isan tale) is one example;
- Explanatory tales are stories about the history or origins of natural things, such as particular animals or places. They refer to people or places that actually existed or are believed to have existed. The tale explains how the person, place, or tradition was conceived. Examples include The Legend of Nong Haan (Sakhon Nakhon province; Mae Mai Island (Chiang Saen district, Chiang Rai); Sa Khu Lu Giant (The origin of the Mekong river); and The Legend of Phaya Khon Khak, which explains the origins of the rocket festival of Isan;
- Animal tales are stories in which the main characters are anthropomorphic animals. One example is The Cunning Rabbit (Cambodia);
- Religious tales are based on the history of local religions, especially Buddhism. The content involves events and stories related to religion;

- Fairytales are stories about magical power, miracles, or the supernatural. Often, the main characters have magical power or magical weapons. Examples include Sin Sai and Campa Si Ton (Laos);
- Jests and jokes are stories about things that would not happen in reality but provide entertainment. Examples include The Blind and the Deaf (Cambodia) and Xieng Mieng (Thailand);
- Other types of tales, such as chain and riddle tales.

(2) Morals are ethical teachings that appear within the tales. They inspire audiences or readers or teach people what they ought to do. Examples include honesty, loyalty, mercy, and forfeit;

(3) Beliefs are the firm thoughts that something is true, often based on religions or superstition. Beliefs appear in folktales, for example, a belief in ghosts and the afterlife appears in Song Phi, Song Phrai, or The Rice Sugarcane Farmer of the Tai Lue. This story affirms traditions related to meritorious behavior; food and things are offered to express love and the desire to send the spirits of the dead onto the next life or to heaven. Examples include the Tan Khun Khao ceremony of the Tai Lue people [32];

(4) Motif can be defined as the smallest element in a folktale having the power to persist in tradition. In order to have this power, it must have something unusual and striking about it. Most motifs fall into three categories: Actors, objects or behaviors of the actors, and place or events in the tales [33] (Stith Thompson, 1977).

### 3.1.2. Context

Context is the data that identify the origin and the setting environments of folktales. There are nine elements as follows:

(1) Creators or different types of tale disseminators. They can be storytellers, monks, other individuals, or organizations that compile and disseminate the tales in different forms;

(2) The title of the tale which may be different based on the languages or areas of origins;

(3) Characters are actors in the tale; they keep folkloric tale going, causing stories and situations to happen. They can be divided into main characters and supporting characters. Examples include the Thao Kumphra Phi Noi (Isan folktale), Thailand and the Frog Princess (Burmese folktale), respectively;

(4) Places are the locations or the scenes that exist within the tales. They can be fictional places, such as the Himmapan Forest, or real ones like Nong Han in the tale of Pha Daeng Nang Ai;

(5) Ethnic groups in tales can be specific to one Mekong Regional tribe or locality. Most tale content treats the legends or history of an ethnic group. For example, the Dao Khon Phi of the Tai Lue group or the Legend of Phra Chao Ok Buad treat the origins of the Tai Lue people, who were there when the one God was ordained;

(6) Languages refers to the various languages used to record and pass down folktales. The language can be a national language, such as Thai, Lao, or Vietnamese or the local language of an ethnic group, such as the Tai Lue language;

(7) Countries refer to the countries in which the folktale originated or was disseminated;

(8) Relations describe the folktale relationship in the region. Folktales disseminated within the countries of the Mekong Region are often shared common cultures or stories among them. Consequently, some stories share similar characters, places, beliefs, or magical items. Between countries, the only difference may be the names of those characters or items, which change to suit the local context or geography. For example, the Tale of Campa Si Ton is found in both Laos and the northeastern region of Thailand. Other tales, such as Twelve Headed Lanka of Xishuangbanna or an adaptation of Ramayana, show the clear influence of India;

(9) Origin is the location where the tale was originally found and taken to be translated and disseminated in a book, journal, database, or video clip published on the Internet.

### 3.1.3. Structure

Structure refers to the media format that is used for folktale's communication. As folktales are considered a type of literature, it can be divided into three following categories based on structure:

(1)   A verbal language form is disseminated through language, either spoken or written, which can be passed down across generations, via storytelling, sermons, or inscriptions on tablets, codices, or palm leaves; it also includes books and modern electronic forms and brochures;

(2)   A nonverbal language form is disseminated by other means of communication, without the use of text or word of mouth.    Examples include objects, statues, and sculptures;

(3)   A mixed form combines both verbal and nonverbal language forms, as in the case of a traditional performance or dance.

### 3.2. Identification of Metadata Requirements

The researcher used the results of the content analysis and related metadata standard to create the guidelines for identifying metadata by adapting and applying the FRBR [24] to analyze the elements of folktales; this was used as a model to design the metadata schema. FRBR model comprised of four elements as follows.

### 3.2.1. Work

Work refers to the way in which the folktale of the Mekong Regional countries was grouped into four overarching categories: type of tales, ethnic groups, morals, and motifs. These data elements are from both the "Content" and "Context" of folktales.

### 3.2.2. Expression

As folktales represent both tangible and intangible cultural heritage, this research classified the elements that expressed folktale knowledge in accordance with [3]'s theory of verbal, nonverbal, and mixed language, which covers all aspects of folktale. These data elements are from the "Structure" of folktales.

### 3.2.3. Manifestation

The manifestation refers to the recorded format of the folktale which can be printed, audio-visual, electronic media, as well as actual items (e.g., paintings, sculptures) and people who have tacit knowledge about the tale. The present study found that folktales within the Mekong region were recorded in many forms, including ancient manuscripts, traditional performances, web databases, and people passing on stories through word of mouth, across generations.

### 3.2.4. Items

The item refers to the existing folktale stored (or targeted for storage) in each different system.

The analysis of the metadata requirements in accordance with the FRBR concept, made it possible to develop the multiple relationships between the elements in Figure 1.

### 3.3. Metadata Schema Development

The metadata schema for the Mekong Region's folktales was developed using the FRBR concept; standard metadata were selected and their appropriateness in the context of the character of folktale knowledge was considered. Next, the researcher selected elements from the existing standards and added new elements to ensure that the developed metadata could describe and enable users to access the information they needed. The metadata also showed the relationship and transference between the various cultures, countries, and ethnic groups.

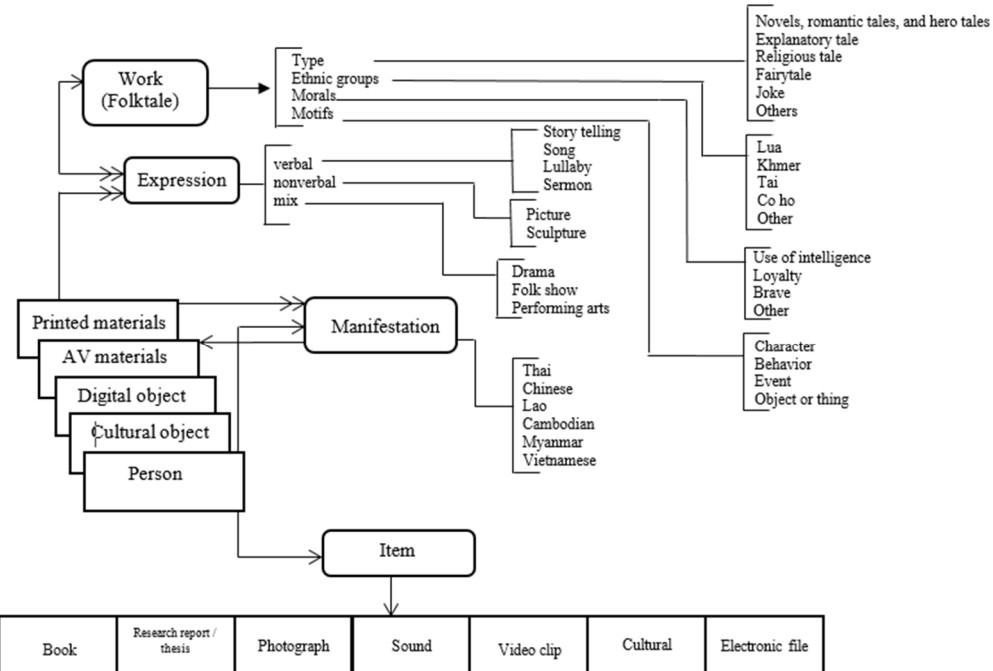

**Figure 1.** Functional requirements for bibliographic records conceptual model of the elements of folktales in Mekong River Basin.

This development of a metadata schema to manage folktale knowledge in the MRB produced the following results. The metadata schema for managing folktale knowledge consists of 18 elements, adapted from the 11 existing standards of the Dublin core metadata element set [27]. These are as follows: identifier, title, creator, contributor, description, relation, language, medium, sources, date, and rights. Seven more elements were added: keyword, character, moral, ethnic group, motif, place, and country. The description of each metadata element consists of a name, definition, format, example, and note, as shown in the following section (Table 1).

**Table 1.** Description of data elements in metadata for folktales in the MRB.

| Element 1 | |
|---|---|
| Name | Identifier. |
| Definition | An identifier code for each folktale stored in the system, use the Roman alphabet, mixed with integral numbers. And URI can also be applied as an identifier for folktale located on the Internet. |
| Format | Two capital letters (Roman alphabet), and four Arabic numbers. |
| Example | TH0001 (For Thai folktale, storage record no. 1). |
| Note | Identifier has two sub-elements:<br>*Sub-element 1*: For folktales in the database collection.<br>    Use two capital letters (Roman alphabet) to form the country code, as signed by ISCO2L.<br>        Thailand = TH<br>        Cambodia = KH<br>        Lao = LA<br>        Vietnam = VN<br>        Myanmar = MM<br>        China = CN<br>    Four Arabic numbers indicate the order of storage records.<br>*Sub element 2*: Use URI syntax<br>    URI = scheme ":" ["//" authority] path ["?" query] ["#" fragment] |



**Table 1.** *Cont.*

| Element 2 | |
|---|---|
| Name | Title |
| Definition | Title of a folktale that appeared in the information object |
| Format | Text as it appeared in the information object, e.g., the title page of a book |
| Example | พญาคันคาก (Praya Kan Kak); ผาแดง นางไอ่ (Pa Daen Nang Ai) |
| Note | Title in the original language |
| | 2.1 The title refers to the title in the original language of the folktale. |
| | 2.2 The parallel title is the title translated into English |
| | 2.3 The uniform/alternative title is any other title that is generally known |

| Element 3 | |
|---|---|
| Name | Creator |
| Definition | Individual or corporate body who created or published the folktale. |
| Format | Text |
| Example | อสิธารา (Asithara) |
| Note | For Thai creators, use "first name and last name" as indicated in the RDA standards. For foreign creators, use "last name, first name." For an anonymous creator, record its appearance |

| Element 4 | |
|---|---|
| Name | Contributor |
| Definition | Individual or corporate body that supported the folktale's collection and publication |
| Format | Text |
| Example | สำนักส่งเสริมวัฒนธรรม มหาวิทยาลัยเชียงใหม่ (Cultural Promotion Office, Chiang Mai University) |
| Note | Supporter of the folktale's collection and publication, e.g., database development, publishing, printing |

| Element 5 | |
|---|---|
| Name | Description |
| Name | Short summary of the folktale story |
| Definition | Text |
| Format | Legend of the origins of Widow Island (Kao Mae Mai) in Chiang Saen District, Chiang Rai Province. In the King Maha Chai Chana period, people caught a taro eel in the Kok river and gave a piece to everyone except one widow. At night there was an earthquake, and all the houses were destroyed except for the widow's house. Since then, the widow's house has been called "Widow Island," and the town has been called "Swamp town" (Vieng Nong Lom) |
| Example | - |
| Note | - |

| Element 6 | |
|---|---|
| Name | Keyword |
| Definition | Words or phrases that represent the key content of folktale, which can involve the concept, object, event, place, person, corporate body, or name in the folktale |
| Format | Text |
| Example | เกาะแม่หม้าย (Widow island); พระเจ้าติโลกราช (King Tilokaraj). |
| Note | - |

**Table 1.** *Cont.*

| Element 7 | |
| --- | --- |
| Name | Character |
| Definition | Name of a character in the folktale |
| Format | Text |
| Example | เซี่ยงเมี่ยง (Siang Miang);<br>เซี่ยงเหมี้ยง (Siang Miang);<br>นางคำกลอง (Nang Kham Klong);<br>ปลาไหลเผือก (Pla Lai Phuek). |
| Note | The name of a character can be recorded in any language used in the folktale. |

| Element 8 | |
| --- | --- |
| Name | Moral. |
| Definition | Teaching words or statements in the folktale. |
| Format | Text. |
| Example | ความซื่อสัตย์ (Honesty);<br>ความภักดี (Loyalty). |
| Note | Teaching words or statements found in the folktale. |

| Element 9 | |
| --- | --- |
| Name | Ethnic group. |
| Definition | Ethnic group(s) found in the folktale. |
| Format | Text. |
| Example | ไทลื้อ (Tai Lue);<br>กะเหรี่ยง (Karen). |
| Note | More than one ethnic group can be recorded---in the Thai language or a dialect, or in English (if appropriate). |

| Element 10 | |
| --- | --- |
| Name | Motif |
| Definition | A dominant characteristic found in a behavior, story, or object in the folktale |
| Format | Text |
| Example | ฉลาดแกมโกง (Cunning) |
| Note | 10.1 Behavior refers to the behavior of the main character, whether positive or negative;<br>10.2 Story refers to a scene or main event in the folktale, e.g., a war or festival;<br>10.3 Object refers to any object found in the folktale, e.g., a weapon or magic tool. |

| Element 11 | |
| --- | --- |
| Name | Place |
| Definition | Geographical name of place or scene found in the folktale |
| Format | Text |
| Example | ป่าหิมพานต์ (Himmapan Forest);<br>หนองหาน (Nong han);<br>เชียงรุ่ง (Chiang Rung);<br>แม่น้ำกก (Kok River). |
| Note | Geographical names can be real or imaginary names |

**Table 1.** *Cont.*

| Element 12 | |
| --- | --- |
| Name | Relation |
| Definition | Relation with other folktale(s) |
| Format | Text |
| Example | Relation.hasVersion ศรีธนญชัย (Sri Thanonchai) |
| Note | Extension of relation (cited from DCMI) |
| | hasVersion          has another version. |
| | isPartOf          is part of a series. |
| | HasPart          forms a part of another series. |
| | isFormatOf          has the same content as another folktale |
| | hasFormat          has the same content as another format |
| | isRelationOf          has content related to another folktale |

| Element 13 | | |
| --- | --- | --- |
| Name | Country | |
| Definition | Country in which the folktale originated or was produced or published | |
| Format | Two-letter country code (Roman alphabet), assigned by ISO2L | |
| Example | Text | |
| | Cambodia = KH | Myanmar = MM |
| Note | China = CN | Thailand = TH |
| | Laos = LA | Vietnam = VN |

| Element 14 | |
| --- | --- |
| Name | Language |
| Definition | Language of the folktale content |
| Format | Two-letter country code (Roman alphabet), assigned by ISO2L |
| Example | Text |
| | Thai = TH |
| | Cambodian = KH |
| Note | Lao = LA |
| | Vietnamese = VN |
| | Myanmar = MM |
| | Chinese = CN |

| Element 15 | |
| --- | --- |
| Name | Medium |
| Definition | Physical format of the media or object used to record or transfer the folktale content |
| Format | Apply MIME type to describe the physical format |
| Example | application/hta |
| | audio/mpeg |
| Note | |

| Element 16 | |
| --- | --- |
| Name | Source |
| Definition | Source of the folktale; this can be the ISBN of a book or the URI of a website |
| Format | Text or |
| Example | Book: ISBN |
| | Website: URI |
| Note | |

**Table 1.** *Cont.*

| Element 17 | |
|---|---|
| Name | Date |
| Definition | Date of publication or production in A.D. |
| Format | YYYY-MM |
| Example | 2020-10 |
| Note | - |

| Element 18 | |
|---|---|
| Name | Rights |
| Definition | Details of copyright or rights for production, reproduction, or publication |
| Format | Text |
| Example | สำนักส่งเสริมวัฒนธรรม มหาวิทยาลัยเชียงใหม่ (Cultural Promotion Office, Chiang Mai University) |
| Note | |

*3.4. Service System Development and Metadata Evaluation*

The present study developed a metadata service system, using the PHP Bootstrap framework to arrange the elements of the designed metadata schema and MySQL to present the search results. Next, the system was evaluated by one folktale expert, two experts on ethnic groups in the Mekong region, and three experts on knowledge organization, metadata, and semantic technology. The evaluation was conducted on 22 June 2021, based on [34]'s continuum of metadata quality, which consists of four aspects: completeness, accuracy, accessibility, and conformance to expectations. The experts were satisfied with the metadata in all four aspects at the highest level (mean values are higher than 3.50), of which the conformance to expectations was satisfied most (mean = 4.83) (Table 2). The experts have also suggested a few points of metadata revision: Adding the element of "Motif," providing features to add new elements for the system users and providing an English version of the metadata (the original was in Thai). The researchers revised the metadata schema in response to the focus group experts to ensure quality improvement.

**Table 2.** Evaluation of the developed metadata.

| Evaluation Items | $\bar{x}$ | S.D. |
|---|---|---|
| **1.    Completeness** | 4.75 | 0.083 |
| Data covers all necessary elements of the folktale as an information object. | 4.75 | 0.463 |
| Data can comprehensively describe the folktale. | 4.63 | 0.518 |
| Data elements can describe any types and formats of the folktale. | 4.88 | 0.354 |
| **2.    Accuracy** | 4.76 | 0.056 |
| The name of each data element is accurate and appropriated. | 4.88 | 0.354 |
| Definition of each data element is clear and accurate. | 4.75 | 0.463 |
| Symbols or abbreviations used in the metadata are easy to understand and accurate. | 4.75 | 0.463 |
| **3.    Accessibility** | 4.67 | 0.278 |
| The use of metadata has helped search for information on folktale that meet the needs. | 4.88 | 0.354 |
| Search options are varied, provide several access to the folktale. | 4.88 | 0.354 |
| Search filters are sufficient and useful. | 4.25 | 0.707 |

**Table 2.** *Cont.*

| Evaluation Items | $\bar{x}$ | S.D. |
|---|---|---|
| **4.     Conformance to expectations** | 4.83 | 0.111 |
| The effectiveness of the search provides the results that conform to the expectation. | 4.75 | 0.463 |
| Metadata can provide benefits for the study of folktale. | 4.75 | 0.463 |
| The system is friendly and easy to use. | 5.00 | 0.000 |

## 4. Discussion

The metadata schema for managing folktale knowledge was developed in accordance with the life cycle of metadata development. The researcher studied the character of the information objects, the scope of the Mekong Region's folktales, the design of the metadata following the FRBR concept [24], and analysis of existing metadata standards. The information object analysis has shown that the folktales of this region have all three components of information objects, whether they are the content of folktales, which is connected, or similar characteristics. This is due to the fact that the countries in the MRB were influenced mainly by either Indian or Chinese culture, as well as Buddhism. Thus, folktales from the Mekong Region have content concerning hell, heaven, doing good deeds, and being rewarded for those good deeds. They also include beliefs about protecting angels, demons, and the afterlife. This finding is in line with research by [32], who found that Tai Lue literature reflected Buddhist values and the Buddhist way of life, influencing even the Tai Lue people's beliefs about nature and the afterlife. This confirms the results of [9], who created a taxonomy of folktale in the Mekong Region and classified the various types of folktale into three main groups: the types, motifs, and origins (GMS) of folktales, of which the terminologies used in this metadata were derived from the taxonomy. Folktales can thus be thought to represent the ideal worlds that tellers wanted to relay or adapt to their own experience, either to influence their audiences to see these worlds and feel proud, or to convey ethical ideas about how to behave, in accordance with the societal values of each locality [35].

The development of the metadata schema for managing folktale knowledge in lline-break the Mekong Region proceeded as per the five existing standards: the DCMI, MODs, VRA Core, CDWA, and performing arts metadata. Each standard has its own limitations regarding describing various information recourses [12,36,37]. However, this metadata development, based on existing standards, will facilitate understanding among different user groups. Thus, it is similar to [38] research, which expanded the potential for accessing an online archive and created The LEADERS schema to describe the archive. That research integrated two existing resources, the encoded archival description, used to describe the collection of archives, and the text-encoding initiative, used to describe actual electronic archives using XML language for creation and access. Integrating metadata standards with existing metadata increases the potential for accessing the online archive. The present study and metadata schema development has used FRBR to analyze elements of the metadata at working levels. As folktales within the Mekong Region are disseminated in different forms, the present analysis of the knowledge structure and structure characters will increase the understanding of the phenomenon of folktale knowledge dissemination, further developing the management of folktales to make it comprehensive. This result is in line with [39], who conducted a survey of the characters in South Korean books, using the FRBR concept to analyze the books, which were viewed as a type of work concept. It is also consistent with [40], who used the FRBR as a frame to develop metadata for elephantology knowledge to analyze the elephantology knowledge of the Kui people, which included many different types of knowledge.

Finally, the system has prepared for web services by using the PHP Bootstrap framework to arrange the elements of the designed metadata schema and MySQL to present the search results. Evaluation of the metadata revealed that it can be used as a tool for

describing and retrieving the folktales stored in the system effectively. The applications of this research are as follows: (1) to use the metadata schema of folktale in MRB to create a digital library or collection to manage the folktale in libraries and organizational collections; and (2) to expand the metadata schema by creating a link with an open-access vocabulary list, such as ontology or a thesaurus, which will support semantic searches and linked open data, enabling people to share resources in their libraries and organizational networks.

**Author Contributions:** Conceptualization, K.K. and K.T.; methodology, K.K. and K.T.; software, W.C. and J.C.; validation, W.C. and T.S.; formal analysis, K.K.; resources, K.K.; data curation, K.K.; writing—original draft preparation, K.K. and K.T.; writing—review and editing, K.T.; visualization, W.C. and J.C.; supervision, T.S.; project administration, K.T. and K.K.; Funding acquisition, K.T. All authors have read and agreed to the published version of the manuscript.

**Funding:** This research received no external funding.

**Institutional Review Board Statement:** Not applicable.

**Informed Consent Statement:** Not applicable.

**Data Availability Statement:** Not applicable.

**Acknowledgments:** This research is supported by the Office of National Higher Education Science Research and Innovation Policy Council, Thailand.

**Conflicts of Interest:** The authors declare no conflict of interest.

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
