# Peer review of "Metadata Schema for Folktales in the Mekong River Basin"

_informatics, doi:10.3390/informatics8040082_

Round 1

Reviewer 1 Report

The paper describes a proposal for a metadata structure designed to represent folklore cultural heritage (CH) from the Mekong Region. The model reuses and adapts elements from the Dublin Core model and defines new elements which aim to reflect the particularities of the case study. A system based on this metadata model for searching and presenting the respective CH items has also been developed and a review by experts has taken place. 

The usefulness of developing an ad hoc metadata structure to capture the particularities of folklore CH of a specific region is not well-justified by the authors. Defining dedicated metadata structures for capturing the particularities of specific regions does not seem like a sustainable and interoperable strategy . Even more so, since  the alignment of the proposed metadata structure with existing established standards seems to suffer from certain weaknesses (see specific comments below), while interrelation with existing relevant vocabularies is not discussed. The use of the terminology could also be clearer.

It is not explained whether and how a metadata structure defined by analysing a particular case study from the Mekong region can be helpful in other cases and can be generalised for representing folklore CH at large.

Moreover, the description of the service system does not provide sufficient information and the evaluation is severely under-developed (see specific comment below).

=================Specific comments throughout the paper==========

"Research has also shown that there are two main types of knowledge management for folklore from the Mekong region. The first is the 
content of folklore [...]. " It is not clear what is the second type.

"Content, which refers to the story, information, or items recorded within the resources. These are latent things that cannot be seen, except through analysis and interpretation [...] ": The term "content" usually refers to a document, image, audio recording etc of a CH item and it is confusing that the authors use this term to refer to "latent things" and information derived through "interpretation" via categories-entities such as "folklore names", "morals", "beliefs" etc.  Similarly,  using the term "structure" to refer to actual verbal, non-verbal or mixed content is quit confusing, e.g. the authors say that an example of what they call "structure" is "content [...] passed down without the use of words, through art, [...]."  (it seems that structure is defined as content).

The relation of the metadata model and the defined categorisations to the already published Ontology of folktales in the Greater Mekong has to be clarified.

"Work refers to the way in which the folklore of the Mekong Regional countries was grouped into four overarching categories: content, ethnic groups, morals, and the motifs.": However, in the previous section morals and motifs are describes a subtypes of content, so the terminology has to be clarified.

A presentation problem exists in page 11.

The definition of  the metadata elements, and particularly those derived from Dublin Core, seem to be ad hoc and not to follow existing standards (in most cases elements are defined as mere strings - no URIs). More specifically:
- For the Identifier element: as stated in the  Dublin Core Metadata Element Set definition, "recommended best practice is to identify the resource by means of a string or number conforming to a formal identification system, such as a Uniform Resource Identifier (URI)". However, the identifiers proposed by the authors do not  have any link to any "formal identification system".
- For the Relation element: as stated in the  Dublin Core Metadata Element Set definition, "Recommended practice is to identify the related resource by means of a URI. If this is not possible or feasible, a string conforming to a formal identification system may be provided." However, the authors seem to treat this element in a way that is not in line with the recommended practice.
- Use of ad hoc textual element Medium instead of the standard use of the Format element which makes use of standards, namely Internet Media Types (MIME). Again, the diversion from recommended practices that require adherence to existing standards is not justified.
- Similar remarks about lack of alignment with standards can be made for other elements, such as rights.

The element "Language" refers to the language of the content, however a field that clarifies the language of the metadata is missing. Moreover, aspects pertaining to the  capabilities of the platform  for searching multilingual items are not discussed.

No information about how the search functionality is implemented and how it relates to the metadata schema is provided (for example, are all elements indexed, what are the search criteria the end user can use, how is multi-linguality treated etc). Figure 2 in particular and the search use case needs to be explained - why are only certain elements mentioned in the dropdown list on the left and how are the elements linked to the free text box on the right, during search?

The evaluation is under-developed. No information is provided as to how the mentioned four dimensions of interest "completeness, accuracy, accessibility, and conformance to expectations" were assessed and what the results have been (just stating that "experts were satisfied" is not enough).

"The applications of this research are as follows: (1) to use the metadata [...] to manage the folklore in libraries and organizational collections; and (2) to expand the metadata schema by creating a link with an open-access vocabulary list [...]": It should be made clear that this is left as future work.

Author Response

Please see in attached file.

Reviewer 2 Report

Try not to conflate "folklore" and "folktale."  In your introduction (lines 39-40), you say that folklore "refers to stories that have been passed down..." a narrower term is actually "oral traditions," including folktales, myths, legends, proverbs, jokes, etc.  FolkLORE is much more broad a label and includes dance, clothing, sayings, customs, etc.  You then use folklore correctly when you describe the three categories (lines 55-60). Wikipedia has a nice overview of the various elements of "folklore."

Line 134: "As most folklore exists in oral forms..." again this is problematic.

Section 3.3.1: the 8 categories do not seem mutually exclusive (i.e., some hero tales will also include miracles and magical powers or have religious overtones; some animal tales may also be jokes).  How discrete do your categories need to be for your taxonomy to "work"?

Line 205: folktales of specific areas are considered "legends."  or do you mean the character names?

Lines 206 and 209: what is the distinction between "morals" and "beliefs"?

Line 214: it's probably best to use Stith Thompson's definition of a motif, since he created the definitive index. Thompson, Stith. 1977 [1946]. The Folktale. University of California Press. ISBN 9780520035379.

Line 241: the "country" of origin, the country of dissemination, and the country of performance (i.e., where it was recorded) may all be different.  Is it important to distinguish these?

Line 251: how do "country" and "origin" differ?

Line 262: would art, fabric, pottery, etc. be included here?  Are you really classifying folkLORE or just oral tales?

Line 280: "traditional performances" is not a recording medium.  The traditional performance had to be captured somehow.  Neither is "word of mouth."  If "manifestation" is the actual medium used to record/preserve the experience, you need to be very careful how you use that term. I'm unclear on the distinction between "manifestation" and "item" as you describe them here. Particularly as your diagram shows "printed materials" as a manifestation, but "book" as an item; AV materials as a manifestation, Video clip as an item.

Line 315: is the ethnic group only those mentioned in the tale, or is it the ethnic group in which the tale originated?

Reviewer 3 Report

at line 133, it is important to notice that tacit knwoledge can also be present within organization`s individuals. SO I would suggest changing the following part from 

"Folklore knowledge is not always recorded or kept within organizations. It can also 133 be held by various individuals; this is called tacit knowledge. " 

to 

"Folklore knowledge is not always recorded or kept within organizations. It can also be held by various individuals in the form of tacit knowledge. " 

metadata would support transforming tatic knowledge into explicit knowledge

for better reading, elements at topico 3.3 could be presented as an apendix

@line 168 "The researcher had previously studied related metadata standards" doesnt sounds like standard english..  (Im not the best person to review this aspectc)

In an overall analisys of the paper, it begins very well structured and fundamented but as it reaches the end,  metadata tests/evaluation and final conclusion could be improved. 

Round 2

Reviewer 1 Report

The authors have addressed most of the comments of the first submission.